# The Combined Effect of the Trapezius Muscle Strengthening and Pectoralis Minor Muscle Stretching on Correcting the Rounded Shoulder Posture and Shoulder Flexion Range of Motion among Young Saudi Females: A Randomized Comparative Study

**DOI:** 10.3390/healthcare11040500

**Published:** 2023-02-08

**Authors:** Shahnaz Hasan, Amir Iqbal, Ahmad H. Alghadir, Asma Alonazi, Danah Alyahya

**Affiliations:** 1Department of Physical Therapy and Health Rehabilitation, College of Applied Medical Sciences, Majmaah University, P.O. Box 66, Al-Majmaah 11952, Saudi Arabia; 2Rehabilitation Research Chair, Department of Rehabilitation Sciences, College of Applied Medical Sciences, King Saud University, P.O. Box 10219, Riyadh 11433, Saudi Arabia

**Keywords:** active stretching, pectoralis muscles, rounded shoulder posture, strength training, trapezius muscle

## Abstract

Background: The shortening of the pectoralis minor muscle (PMi-M) and weakening of the lower trapezius muscle (LTr-M) affect scapular movement, resulting in the development of a rounded shoulder posture and reduction in the shoulder flexion range of motion (SFROM). Objective: This study evaluated the combined effect of LTr-M strengthening and PMi-M stretching on correcting the rounded shoulder postures and SFROM among young Saudi females. Methods: This study was based on a two-arm parallel-group repeated measures randomized comparative design. A total of sixty female participants with rounded shoulder postures were recruited and randomly allocated into groups 1 and 2 (*n* = 30/group). Each group performed supervised PMi-M stretching; however, group 2 performed a combination of LTr-M strengthening and PMi-M stretching. The outcomes, including rounded shoulder posture and SFROM, were assessed using the pectoralis minor length test (PMLT) and universal goniometer. A repeated measure ANOVA was used to compare the differences within-group and between-group for the outcomes measures at one-week (baseline) pre-intervention, two weeks, and three -weeks post-intervention. The significance level was set at q > 2.00 and *p* < 0.05 for all respective statistical analyses. Results: The within-group comparison revealed significant improvements (q > 2.00) in the outcomes of PMLT and SFROM when comparing their post-intervention scores to the baseline scores. The between-group comparison revealed a significant and an insignificant (q < 2.00) difference in the outcomes of PMLT and SFROM, respectively when comparing their scores at the second- and third-week post-intervention. Furthermore, the effect size of the intervention suggests an advantage of group 2 over group 1 in increasing the resting length of the PMi-M only among young Saudi females. Conclusions: The combined effect of LTr-M strengthening and PMi-M stretching was more beneficial than PMi-M stretching alone in correcting the rounded shoulder posture among young Saudi females by increasing PMi-M resting length. However, it could not yield a differential improvement in the SFROM outcome among them.

## 1. Introduction

Rounded shoulder posture appears to be one of the most common musculoskeletal anomalies of the shoulder complex resulting in shoulder joint pain in patients being most prevalent clinically, with one in three people experiencing it in their lives [1,2]. Rounded shoulder posture is a typical maladaptive posture that increases due to repetitive work and poor posture [3], and occurs in up to 73% of the group of healthy participants between the ages of 20 and 35 years [3,4,5,6]. It is characterized by a protracted, downwardly rotated, and anteriorly tipped scapula position with increased cervical lordosis and upper thoracic kyphosis [3]. In the literature, rounded shoulder posture is described as abduction, the elevation of the scapula giving an appearance of a hollow chest [7,8]. The rounded shoulder posture is associated with tightness of the serratus anterior, pectoralis minor, pectoralis major, and upper trapezius muscle and weakness of the middle and lower trapezius [5]. The origin of the pectoralis minor (PM) muscle is from the third, fourth, and fifth ribs near the sternocostal junction and insertion into the coracoid process of the scapula [9]. Their function is to increase scapular posterior tipping and decrease internal scapular rotation during arm elevation [10,11]. The function of the lower trapezius is an essential component of normal scapulohumeral rhythm [12,13]. Maintaining the rounded shoulder posture and performing repetitive scapular movements can result in adoptive shortness of the pectoralis muscle. This shortness is a potential mechanism for shoulder and neck pain [14]. The pectoralis minor has also been recognized as a muscle requiring stretching in individuals with rounded shoulder posture [10,15] or with clinical shoulder impingement [16,17,18,19]).

Adoptive shortness of the pectoralis minor would not demonstrate normal flexibility [20]. More precisely, it would present less total excursion than a relatively longer muscle because the number of sarcomeres decreases in series, and fewer actin–myosin cross-bridges to uncouple [21]. Therefore, an adaptively shorter pectoralis minor muscle does not allow the scapula to rotate fully upward, externally rotate, posteriorly tip, or elevate [11,22].

Clinicians hypothesized that strengthening posterior scapular stabilizers combined with stretching of the pectoral muscles can correct muscle imbalance, rounded shoulder posture, and normalize the scapulohumeral rhythm [23,24]. We hypothesized that because of muscle adaptively shortening due to repetitive movements and being unable to contract in the optimal position, muscle strength would decrease and cause an imbalance between agonist and antagonist muscle strength of the shoulder joint and alter scapular kinematics due to pectoralis minor shortness, which is similar to the shoulder impingement. In addition, different professions, cultural customs, practices, living and eating styles, and habits of different geographical areas affect people’s health, including posture and postural alignments, especially in females, such as rounded shoulder posture.

To date, no studies have used such a combination of intervention approaches, including the LTr-M strengthening and the PMi-M stretching for correction of the rounded shoulder posture and improvement of the limited SFROM by increasing the resting length of the PMi-M among young Saudi females. Therefore, this study aimed to determine the effect of LTr-M strengthening and PMi-M stretching on correcting the rounded shoulder posture and SFROM among young Saudi females. This study will provide insight for global physiotherapists, including the Arab countries opting for the best possible, direct, accurate, and evidence-based intervention approach (single/combined) for correcting rounded shoulder posture and improving limited SFROM among young Saudi females. In addition, adopting this specific add-on-intervention approach will increase the accuracy and speed of prognosis and ease the effort for selecting and applying the intervention’s approach while aiming to treat according to the goals of the intervention plan.

## 2. Materials and Methods

### 2.1. Study Design

The study followed a two-arm parallel-group repeated measures randomized comparative design. The effect of the combination of interventions on the outcomes was evaluated among young females with rounded shoulders in Saudi Arabia.

### 2.2. Study Sample Size

Computer software, G*Power 3.1.9.4, was used to estimate the effective sample size for this study. A computer priori t-test (two-tailed) analyzed the outcome scores of the pectoralis minus length test (PMLT) of a sample of six participants at the pilot stage, keeping a value of power 0.80 (80%), mean differences 0.83, standard deviation 0.34, level of significance *p* < 0.05, and effect size d- 0.853; obtained a sample of 46 (23/group) participants in this study. With the assumption of 20% attrition in sample collection, a total of 58 participants was required to satisfy the power sample.

### 2.3. Ethical Consideration

The study obtained ethical clearance from the ethics sub-committee at King Saud University (file Id: RRC-2019-21, dated: 26 September 2019) before the experiment was started, and this was conducted in accordance with the principles outlined in the Declaration of Helsinki (2010). The study trial was registered on the “ClinicalTrial.gov Protocol Registration and Results System (clinicalTrial.gov ID: NCT04686123; dated: 24 December 2020). Before the beginning of the study, each participant had returned a signed informed consent for participation in this study.

### 2.4. Study Setting

The participants for this study were approached using pamphlet distribution and banners inside and outside the outpatient physiotherapy department of our university hospital. Sixty healthy young female participants were recruited in the study within 6 months, starting from 29 September 2019, and completed on 27 February 2020. 

### 2.5. Participants

A total of sixty young female participants with rounded shoulder postures were recruited who met the inclusion criteria in the study. The inclusion criteria were set as follows: the participants aged between 18 and 25 years, with no history of shoulder trauma, current shoulder pathology, thoracic scoliosis, and kyphosis deformity. The participants were excluded from the study if they received any other form of medical treatment or with thoracic scoliosis or kyphosis deformity. The participants were confirmed to have rounded shoulder posture by a senior physiotherapist with more than ten years of experience through an examination in which they were instructed to lie down in a supine position resulting in a measurement that exceeded 2.5 cm between the table’s surface and the posterior surface of their shoulder peaks [25,26,27]. 

### 2.6. Procedures

A total of sixty female participants with rounded shoulders were approached in the O.P.D, the physiotherapy department of our university hospital, and recruited for the study. Before participation, each participant read and signed an informed consent form. The participants’ demographic data were recorded and randomized into two groups, group 1 (PMi-M stretching only) and group 2 (PMi-M stretching combined with LTr-M strengthening), with 30 participants in each group. The randomization scheme was generated using the website Randomization.com ⟨http://www.randomization.com (accessed on 11 October 2019)⟩ [28]. The examiner and statistician were blinded to the participant’s group allocation. The schematic presentation of study procedures has been explained in Figure 1 using a CONSORT 2010 flow diagram. 

### 2.7. Outcome Measures

The rounded shoulder posture and shoulder flexion range of motion were measured by the pectoralis minor length test (PMLT) and a universal goniometer, respectively. Both of these methods have been found to have acceptable reliability [11,22]. An assessor who was kept blind to the study protocol recorded all the outcome measures at baseline, two weeks, and 3 weeks post-intervention.

The participants performed a brief warm-up involving three repetitions of active shoulder movement before measurements. The pectoralis minor length test (PMLT) is the vertical distance between the posterior surface of the shoulder peak and the supporting surface with the participant in supine lying with elbows flexed and arm by the side of the body (Figure 2a) [21]. A participant with a distance of more than 2.5 cm is considered to have short pectoralis minor (rounder shoulder posture).

An active shoulder flexion range of motion (SFROM) was measured in a seated position on a chair with the back supported and the trunk upright using a universal full circle goniometer (National 360 Goniometer, National Tools Ltd., Shop No.-1460, Daryaganj, New Delhi, India) (Figure 2b,c). The goniometer was positioned with the fulcrum at the glenohumeral joint’s midpoint, the stable arm parallel to the mid-line to the side of the thorax, and the moving arm parallel to the longitudinal axis of the arm bone, i.e., humerus. Before taking measurements, the therapist moves the participant’s arm passively toward the ceiling to the complete end range for one repetition to familiarize the participant with the shoulder flexion motion. After a passive shoulder flexion by the therapist, participants were asked to actively move through the motion to the end range. Verbal and manual cues were used to correct shoulder flexion motion. Once the participant reached the end range, the therapist started the measurement. After completion of the measurements, the participant was asked to return the arm to the side. From the resting position, the participant was then asked to raise the arm again in the scapular plane to the complete end range. The therapist repeated goniometric measurements for two measurements with 5-min rest, and the average score was documented for data analysis. 

### 2.8. Interventions

All the participants performed the LTr-M active strengthening and PMi-M active stretching per their allocated intervention protocols under the supervision of a specialist physiotherapist with more than ten years of experience at the O.P.D physiotherapy unit of our university hospital. Before measurement, all participants underwent a warm-up session consisting of three repetitions of full, active shoulder range of motion, such as abduction, flexion, extension, and rotations. After a warm-up session, the participants from both groups performed a common PMi-M active stretching while the participants from group 2 performed an additional LTr-M active strengthening. All the participants performed their stipulated intervention protocol for both shoulders; however, the average outcomes scores for the dominant shoulder were taken for analysis. 

#### 2.8.1. Group 1 (PMi-M Active Stretching Only)

Each participant performed an active stretching exercise to elongate the PMi-M length. The participant performed PMi-M active stretching in a standing position with the shoulder abducted at a 90°, elbow flexed at a 90°, and palm placed on a flat planar surface. The participant then rotated the trunk away from the elevated arm, increasing the horizontal abduction at the shoulder [22], maximizing the stretch across the chest, and held this position for 30 s. The participants performed two sets of three-stretches per session per day for five consecutive days a week for three weeks. The one-minute gap was kept between two sets of stretches.

#### 2.8.2. Group 2 (PMi-M Active Stretching Combined with LTr-M Active Strengthening)

First, the participants performed PMi-M active stretching in a similar manner and dosage as in group 1. After 5 min of rest, the active strengthening exercise was performed to increase the LTr-M strength. The LTr-M strength was performed by each participant in prone lying with arms placed diagonally overhead in line with the lower fibers of the trapezius and shoulder externally rotated. The participant then raised the arms overhead in line with the LTr-M fibers, causing depression and adduction of the scapula. This position was maintained for 30 s. The exercise was performed in 2 sets of 5 repetitions per session per day for five consecutive days in the first week and progressed to 10 and 20 repetitions per session for the second and third weeks, respectively. The one-minute gap was kept between two sets of exercises.

### 2.9. Data Analysis

The Shapiro–Wilk test of normality was used to analyze the distribution of the baseline scores for the demographic and outcomes variables, including age (years), height (cm), weight (kg), rounded shoulder posture as pectoralis minor length test (PMLT), and shoulder flexion range of motion (SFROM) (°). The statistical analyses within-group and between-group were performed using the GraphPad Prism version 6, with a ‘q’ value greater than two (q > 2.00) considered significant. Repeated Measure ANOVA (Tukey’s multiple comparison test) was used to compare the differences within the groups and between the groups for the outcome measures at baseline, two weeks, and 3 weeks post-intervention. Furthermore, Cohen’s *d* test was used to see the actual effect size of the intervention over outcomes between the groups. In addition, Pearson’s correlation coefficient (r) was used to detect the association between the rounded shoulder posture (PMLT) and SFROM among the participants (N = 60).

## 3. Results

Sixty out of seventy-eight female participants were recruited in this study. Nine participants did not satisfy the inclusion criteria, five were denied participation due to their unavailability for three weeks, and four left the study without any reason. All variables’ data were distributed homogeneously in groups 1 and 2 except for the variable age in group 2 (statistic: 0.840; *p* = 0.000). Furthermore, the mean and standard deviation (SD) for demographic details, including age, height, and weight, along with the baseline values of outcome measures (PMLT and SFROM) of group 1 and group 2, are explained in Table 1. The outcomes scores of PMLT and SFROM at baseline, two weeks, and three weeks post-interventions are represented in tables as PMLT1, PMLT2, and PMLT3 and SFROM1, SFROM 2, and SFROM3, respectively. 

Within-group analysis for the variables PMLT and SFROM showed a significant improvement (q > 2.00) when comparing their post-intervention scores at different time intervals with the baseline scores among both groups (1 and 2), as presented in Table 2.

In group 1, the PMLT scores showed a significant mean difference (∆M) when baseline scores were compared with the post-intervention scores at two time points, such as PMLT 1- PMLT2 (∆M = 1.251; q = 7.12), PMLT1–PMLT3 (∆M = 2.413; q = 13.75), PMLT2- PMLT3 (∆M = 1.162; q = 6.619). Similarly, in group 2, the PMLT scores showed a significant mean difference (∆M) when comparing the baseline scores to the post-intervention scores at two time points, such as PMLT1–PMLT2 (∆M = 1.882; q = 10.72), PMLT1–PMLT3 (∆M = 3.031; q = 17.27), PMLT2–PMLT3 (∆M = 1.149; q = 6.546). Moreover, in group 1, the variable SFROM showed a significant mean difference (∆M) when the baseline was compared with the post-intervention scores at two time points, such as SFROM1- SFROM2 (∆M = 4.233; q = 5.233), SFROM1–SFROM3 (∆M= 9.767; q = 12.07) SFROM2- SFROM3 (∆M = 5.533; q = 6.84). In addition, in group 2, the SFROM scores showed a significant mean difference (∆M) when comparing the baseline scores to the post-intervention scores at two time points, such as SFROM1- SFROM2 (∆M = 5.333; q = 6.592), SFROM1- SFROM3 (∆M = 10.3; q = 12.73), SFROM2- SFROM3 (∆M = 4.967; q = 6.139).

However, between-group (1 vs. 2) analysis showed a significant mean difference (∆M) for the variables PMLT when compared at post-intervention time points, such as 1PMLT2–2 PMLT 2 (∆M = 0.889; q = 5.064), 1 PMLT3–2 PMLT3 (∆M = 0.8763; q = 4.991). In contrast, the variable SFROM presented a non-significant mean difference between groups 1 and 2 at post-intervention time points, as described in Table 3 and Figure 3 and Figure 4.

Furthermore, the Cohen’s *d* test revealed a progressive increment in the treatment effect size for the variable PMLT when compared to the d-value between groups 1 and 2 at two weeks (*d* = 0.913) and three weeks (*d* = 1.278) post-intervention, however, a very small effect size was detected for the variables of SFROM when compared between the groups at two weeks (*d* = 0.106) and three weeks (*d* = 0.026) post-intervention, as described in Table 3.

In addition, Pearson’s correlation coefficient (*r*) detected the strength of association between the variables and revealed the weak correlation (95% CI) between a PMi-M length test (rounded shoulder posture) and SFROM at different time intervals of the study such as at baseline (*r* (58) = 0.133; *p* = 0.311), two weeks (*r* (58) = 0.002; *p* = 0.986), and three weeks (*r* (58) = 0.258; *p* = 0.047) post-intervention (Figure 5). 

## 4. Discussion

This study aimed to evaluate the effect of LTr-M strengthening in combination with PMi-M stretching to correct rounded shoulder posture and SFROM among young Saudi females by observing the changes in PMi-M length. Several clinicians postulated that the strengthening scapular stabilizers combined with the stretching of pectoral muscles can correct posture and muscle imbalance and alter the scapulohumeral rhythm. The stretching and strengthening exercise programs are used clinically to correct the muscle imbalances between the anterior and posterior shoulder musculature [19,29]. The result obtained in this study was a tremendous and statistically significant difference between the groups, supporting the hypothesis for the correction of the rounded shoulder posture. 

One previous study compared three techniques and concluded that the unilateral self-stretch was better than the supine or sitting manual stretch techniques for increasing the distance between the origin and insertion of the PM [9]. Decreased muscle length may lead to the loss of extensibility due to the decreased number of sarcomeres in series and fewer actin–myosin cross-bridges [20], the type of titin protein, and the shortening of connective tissue [30]. Strengthening the weakened muscles leads to biomechanical movement and obtaining of the appropriate direction of abnormal parts. Indeed, stretching the hypertrophied muscle (shortened) and strengthening the weakened muscles considerably improves the rounded shoulder abnormality [31]. Stretching has long been applied to improve ROM and flexibility measures, but it can also significantly negatively affect neuromuscular performance. Stretching places strain on the origin and insertion of the muscle and may cause damage to the sarcomeres. From this evidence, it can be hypothesized that since there were no deficits in muscular performance, strengthening showed better results than stretching with strengthening improving flexibility and ROM. Autopsy studies have shown that the lower trapezius predominantly contains type-1 fibers (76%) [32].

In trapezius rehabilitation, its postural function is addressed and retrained. Hence, in this study, the strengthening intervention protocol for lower trapezius was progressive by increasing the number of repetitions from five repetitions in the first week to ten repetitions in the second week and twenty repetitions in the third week. We also took shoulder flexion range of motion in the sagittal plane as an outcome measure to assess the pectoralis minor muscle length for two reasons. First, the tight pectoralis minor muscle and slouch posture were discovered to relate to a decrease in posterior tilting and upward rotation of the scapula [33], and these motions are essential for completing a full shoulder flexion range of motion [34,35].

Secondly, the investigator conducted a pilot study and found that the participants with short pectoralis minor muscles also demonstrated a limited shoulder flexion range of motion in the sagittal plane. However, no study has been reported investigating the effect of minor pectoralis muscle length on shoulder flexion range of motion in the sagittal plane.

The results of this investigation support the premise that pectoralis minor muscle active stretching with added lower trapezius muscle strengthening can be more effective than the pectoralis minor muscle active stretching alone in correcting the rounded shoulder posture by increasing the resting length of PMi-M.

Comparing the mean pectoralis minor length test (PMLT) scores for each intervention group revealed a significant difference (q > 2.00) at the end of the session. The trend of the mean scores during the study demonstrated a progressive decrease in mean PMLT scores in both the intervention groups, suggesting that both the intervention protocols were effective in increasing the length of PMi-M, which lead to the correction of a rounded shoulder posture. The interaction of group and time was also significant, meaning that mean PMLT scores for the groups were different at the end of the study. Thus, both intervention protocols were found to be effective in increasing the length of PMi-M. This finding is consistent with a study that reported that three weeks of active stretching exercises for a short pectoralis minor significantly increased its length [3].

Comparing the mean PMLT scores of both groups at two and three weeks after the intervention session revealed a significant difference between the groups (q > 2.00). It indicates that the intervention protocol, including LTr-M strengthening and PMi-M stretching, was superior to the intervention protocol, including the PMi-M stretching alone, in increasing the length of the PMi-M. In addition, Cohen’s *d-*test for the effect size of intervention protocols on PMLT also revealed a very large effect size between the groups when compared at two weeks (*d* = 0.913) and three weeks (*d* = 1.278) post-intervention. Thus, the LTr-M active strengthening with PMi-M active stretching is more beneficial than PMi-M active stretching alone in correcting the rounded shoulder posture by increasing the resting length of the PMi-M. This also suggested that the add-on effect through the LTr-M strengthening was proven effective in sustaining the increased length of PMi-M, leading to the correction of the rounded shoulder posture.

The rank-sum of mean SFROM scores revealed a significant difference (q > 2.00) in SFROM scores when comparing pre-and post-intervention scores within each group. Compared with the baseline scores, both groups showed a significant improvement in SFROM scores at the end of the third-week post-intervention session, suggesting that both intervention protocols effectively increased SFROM among young females with rounded shoulder postures. In contrast, Tukey’s multiple comparison tests indicated no differences for the variable SFROM when comparing the mean SFROM scores between the groups. Moreover, Cohen’s d-test also reported a small effect size of intervention protocol between the groups at two weeks (*d* = 0.11) and three weeks (*d* = 0.03) post-intervention Thus, the intervention protocol, including LTr-M strengthening and PMi-M stretching, is equally as effective as the intervention protocol, including the PMi-M stretching alone in improving the SFROM among young females with rounded shoulder posture. It also suggested that the combination of LTr-M strengthening with PMi-M stretching did not yield an additive effect in improving the SFROM.

The findings of this study are consistent with the results of a previous study that found the pectoralis minor active stretching, with added lower trapezius strengthening exercise program, was more effective than active stretching alone in increasing the PMi-M length [3]; however, the authors did not investigate the effect of the exercise program on SFROM. In another study, investigators reported no significant change in resting scapular position following an active and moderately aggressive strengthening exercise program for scapular retractors and pectoral stretching in participants with rounded shoulder postures. However, significant changes in scapular kinematics were found during arm elevations after practicing shoulder-shrugging and scapular retraction with shoulder 90° abducted exercises as strengthening exercises and using corner-stretch exercises for pectoral stretching [19]. In contrast, one study found that stretching with soft tissue mobilization of the pectoralis minor significantly reduced forward shoulder posture [3].

The study was limited to using only two-dimensional scapular motions in the strengthening program; however, PMi tightness affects the three-dimensional motion of the scapula [10]. A three-dimensional motion could use an adjunct such as EMG Biofeedback in strengthening exercises to ensure appropriate activation of the desired muscle. Other scapular retractor muscles could be strengthened along with the lower trapezius muscle to correct muscle imbalance [10,35]. Posture and muscle imbalance are associated as a part of the pathological process. The concept of correcting posture and its associated muscle imbalance through stretching and strengthening programs has been accepted clinically [36,37]. However, the evidence to support these theories is limited, with researchers reporting equivocal findings [38,39]. In contrast, few researchers found a poor correlation between the scapular resting position, pectoralis minor, and middle trapezius muscle force [40].

Future investigations should address the effect of other PM stretch techniques on PM length, shoulder function, and pain and adding strengthening exercises for other posterior scapular stabilizers to pectoralis minor muscle stretching exercises in increasing pectoralis minor muscle length. Future studies should also assess the effectiveness of interventions in male patients with shoulder pathology.

## 5. Conclusions

Therefore, the present study concluded that LTr-M strengthening and PMi-M stretching are more beneficial than PMi-M stretching alone in correcting rounded shoulder posture among young Saudi females. This beneficial effect was achieved due to sustaining the gained length of PMi-M by strengthening the LTr-M simultaneously. There was a significant increase in PMLT in all participants but not in SFROM since no correlation existed between PMLT and SFROM, possibly due to the shortened duration of the strengthening protocol. Thus, physiotherapists around the globe, including in Arab countries, should adopt a combined intervention approach, including LTr-M strengthening and PMi-M stretching, aiming to correct the rounded shoulder posture.

## Figures and Tables

**Figure 1 healthcare-11-00500-f001:**
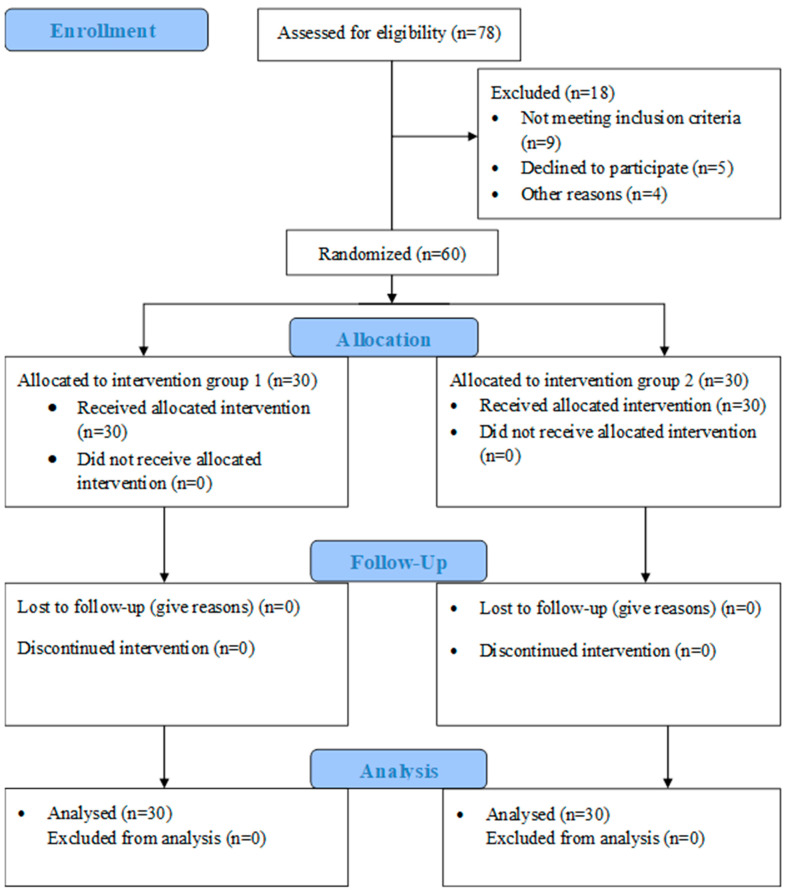
A CONSORT (2010) flow diagram presents the procedures, including study enrollment, allocation, follow-up, and analysis.

**Figure 2 healthcare-11-00500-f002:**
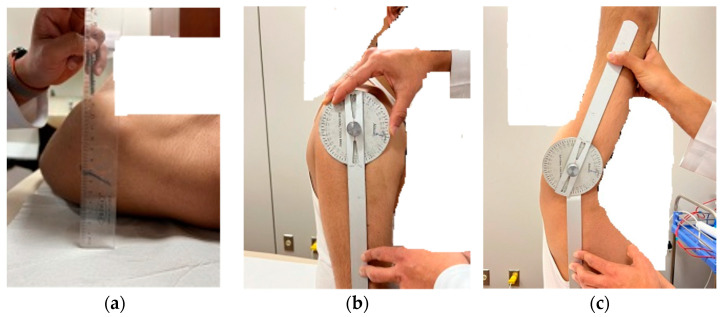
Showing the assessments for the rounded shoulder/PMLT (**a**), SFROM initial (**b**), and end (**c**) position.

**Figure 3 healthcare-11-00500-f003:**
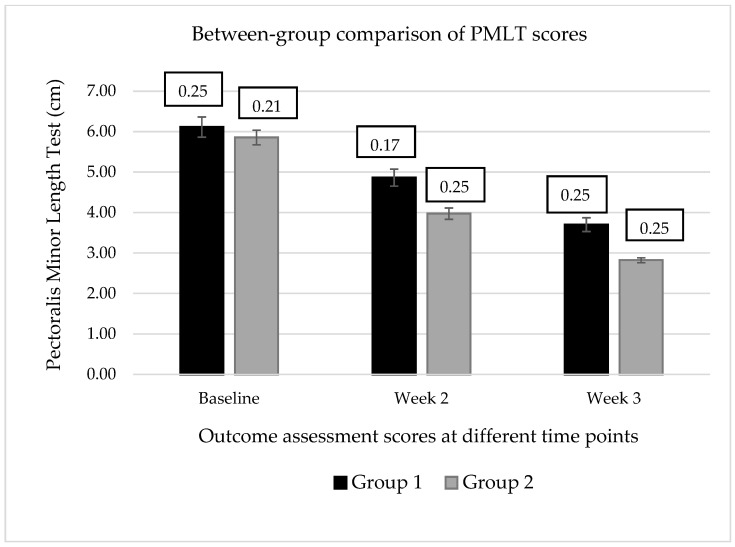
Comparison of the means ± standard deviation for the dominant pectoralis minor length test (PMLT) between groups.

**Figure 4 healthcare-11-00500-f004:**
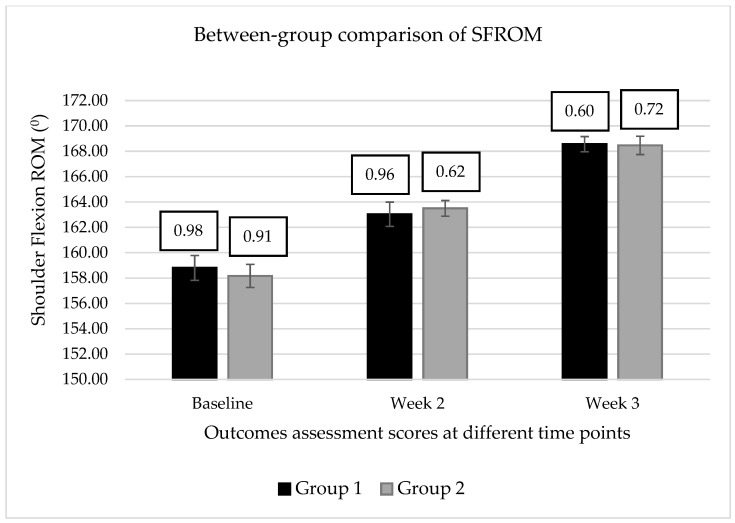
Comparison of the means ± standard error for the dominant shoulder flexion range of motion (SFROM) between the groups.

**Figure 5 healthcare-11-00500-f005:**
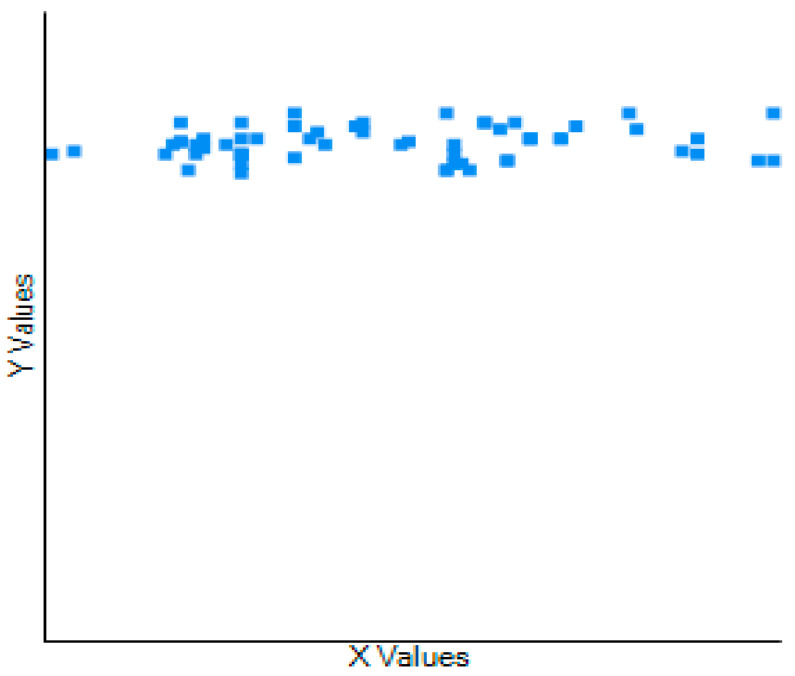
The correlation between the participants’ PMLT (X values) and SFROM (Y values) scores at baseline.

**Table 1 healthcare-11-00500-t001:** Baseline measures (mean and standard deviation) and test of normality for sample distribution (Shapiro–Wilk test of normality).

Variables	Baseline Value and Shapiro–Wilk Test of Normality					
	Group 1 (*n* = 30)			Group 2 (*n* = 30)		
	Mean ± SD	Statistics	*p*-Value	Mean ± SD	Statistics	*p*-Value
Age (Yrs.)	22.83 ± 1.49	0.933	0.060	22.53 ± 1.11	0.840	0.000 *
Height (cm)	174 ± 0.04	0.972	0.585	173 ± 0.05	0.979	0.791
Weight (kg)	63.73 ± 5.25	0.962	0.344	62.70 ± 4.83	0.946	0.130
PMLT (cm)	6.11 ± 1.37	0.939	0.088	5.86 ± 1.00	0.952	0.194
SFROM (°)	158.80 ± 5.37	0.952	0.188	158.17 ± 4. 96	0.964	0.398

SD: Standard deviation; *—Significant value if *p* < 0.05; *n* = Sample size of each group.

**Table 2 healthcare-11-00500-t002:** Pair-wise comparison of the variables means differences within-group. (Repeated Measures ANOVA test: Tukey’s multiple comparisons test).

Variables (Pairwise)	Group 1		Group 2	
	∆MD	q-Value	∆MD	q-Value
PMLT1-PMLT2	1.251	7.127 *	1.882	10.72 *
PMLT1-PMLT3	2.413	13.75 *	3.031	17.27 *
PMLT2- PMLT3	1.162	6.619 *	1.149	6.546 *
SFROM1-SFROM 2	−4.233	5.233 *	5.333	6.592 *
SFROM1-SFROM 3	−9.767	12.07 *	10.3	12.73 *
SFROM2-SFROM 3	5.533	6.84 *	4.967	6.139 *

PMLT: Pectoralis Minor Length Test; SFROM: Shoulder Flexion Range of Motion; *—Significant value if q > 2.00; ∆MD: Mean difference scores.

**Table 3 healthcare-11-00500-t003:** Pair-wise comparison of the variables’ mean differences and intervention effect size between the groups using Tukey’s multiple comparison test.

Variables (Pairwise)	∆MD	ANOVA		Cohen’s *d*
		DF	q-Value	
1PMLT2–2 PMLT2	0.889	174	5.064 *	0.913 ˆ
1PMLT3–2 PMLT3	0.876	174	4.991 *	1.278 ˆˆ
1SFROM2–2SFROM 2	−0.467	174	0.577 ^ns^	0.106
1SFROM3–2SFROM3	0.1000	174	0.124 ^ns^	0.026

PMLT: Pectoralis Minor Length Test; SFROM: Shoulder Flexion Range of Motion; *—Significant value if q > 2.00; ns—Non-significant value if q < 2.00; **ˆ**—Large effect-size if d-value between 0.8 and 1.0; **ˆˆ**—Very large effect size if d > 1.0; ∆MD: Mean difference scores.

## Data Availability

The data associated with the paper are available from the corresponding author upon reasonable request.

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
