# Peer review of "The Combined Effect of the Trapezius Muscle Strengthening and Pectoralis Minor Muscle Stretching on Correcting the Rounded Shoulder Posture and Shoulder Flexion Range of Motion among Young Saudi Females: A Randomized Comparative Study"

_healthcare, 2023, doi:10.3390/healthcare11040500_

Round 1

Reviewer 1 Report

The aim of the study was to evaluate the effect of therapy consisting in strengthening the lower trapezius muscle and stretching the pectoralis minor muscle on correcting the posture of rounded shoulders and the range of flexion of the shoulder joint in 60 young Saudi women. The study included a division into two research groups. In group 1 (N=30), only pectoralis minor muscle stretching therapy was used, while participants in group 2 (N=30) received a combination of pectoralis minor muscle stretching and lower trapezius muscle strengthening. A universal goniometer and pectoralis minor length test (PMLT) were used to compare the effects of these two treatments. Measurements were taken at three time intervals: 1st-week (baseline) pre-intervention, 2nd, and 3rd-weeks post-intervention.

At the beginning, I would like to congratulate the authors of the idea and the correct design of the study itself. I believe that scientific research with a clear application purpose provide valuable data and conclusions, which makes them very important for broadly understood clinical practice.

Although the manuscript is interesting and generally well written, I do have a few comments and questions.

In the subsection "2.5 Participants", the authors describe the qualifications of the participants for the study. I am interested in who carried out the rounded shoulders diagnostic procedure? (physiotherapist, doctor, researcher?) What was his professional experience?

Error in citation notation - missing comma (line 127).

In Figure 1, presenting a CONSORT flow diagram is a very important element in the case of randomized clinical trials. I believe that at the "Allocation" level, the authors should sign what was the name of the group/intervention. This applies to the left and right arms of the diagram. This will help the reader better understand the flow diagram.

In subsection '2.7. Outcome measures” the authors described the research methodology too vaguely. Was the measurement of the PMLT test performed immediately after the participant assumed the supine position? Did you wait some time before measuring? What exactly was the goniometer (brand, model)? What was the point of application of the goniometer? Were one measurement of movable property carried out, or were several measurements performed and the average calculated? Was it an active or passive ROM? Was the maximum range of motion tested or until pain/discomfort occurred? This subsection should also include information on whether one or both sides of the body were assessed.

I think it would be very helpful to add a figure showing how the measurements were carried out.

What was the professional experience (years) of the specialist responsible for the correct application of the intervention?

In subsections 2.8.1. and 2.8.2. there should be information whether the intervention concerned one or both sides of the body.

Figure 2. - applies to the right or left muscle? Is it the average of the right and left? I think this information should be included in the title of the figure. The same applies to the other tables and figures.

To the paragraph (lines 254-258) - adding a figure showing the correlation diagram would enrich the results section.

I think the discussion section should be improved. I have the impression that the written discussion is mostly a description of the study and a repetition of the most important results. There is no explanation of the mechanisms responsible for the observed significant statistical differences or their absence (the lack of recorded statistical differences also proves something). An important aspect of the discussion concerning the comparison of the obtained results to the results/conclusions of the works of other authors in this scientific field was definitely limited. In the discussion paragraph (lines 324-335) there is a slight comparison with works on similar subjects. Even if there is a small number of similar studies, the authors should try to compare their own results with other works in the area of stretching and strengthening the muscles responsible for shoulder mobility and correct posture. These can be works related to manual therapy, METs or other rehabilitation programs aimed at increasing the mobility of the shoulder joint or affecting posture, movement patterns or the broadly understood function of the locomotor system during everyday activities. When comparing the results/conclusions of other studies, one should also consider explaining these differences (if any). Such a comprehensive summary gives a full view of how the evaluated therapy compares with other (more or less similar) therapeutic methods. In addition, it will strengthen the bibliography.

At the end of the discussion section, a paragraph describing the limitations of the study and the authors' suggestions for the direction of future research in this particular scientific area are missing.

Section "5. Conclusions' should be corrected. Conclusions should not only be a repetition of the most important results, but an attempt to explain these differences/changes (or lack thereof).

The authors wrote: "Therefore, the study concluded that the combined effect of LTr-M strengthening and PMi-M stretching is more beneficial than PMi-M stretching alone in correcting rounded shoulder posture among Saudi young females." Why? What do the authors think contributed to this difference? There is no explanation here.

The authors conclude: "However, it couldn't yield a differential improvement in the outcome of SFROM among the same participants." Here should be a brief explanation of this result. Why, according to the authors, was there no difference? Is it a matter of the therapeutic techniques used? Is the duration of therapy? Maybe the frequency of application of treatments had an influence on it?

Author Response

The authors attached a point-by-point response letter to the reviewer's 1, for your perusal. Kindly, do the needful.

regards

Author Response

The authors attached a point-by-point response letter to the reviewer's 2, for your perusal. Kindly, do the needful.

Regards

Round 2

Reviewer 1 Report

I congratulate the authors for improving their manuscript.

Reviewer 2 Report

I have confirmed that the points pointed out have been improved.